# Strategies to Reduce Post-Hemorrhoidectomy Pain: A Systematic Review

**DOI:** 10.3390/medicina58030418

**Published:** 2022-03-12

**Authors:** Varut Lohsiriwat, Romyen Jitmungngan

**Affiliations:** 1Division of Colon and Rectal Surgery, Department of Surgery, Faculty of Medicine Siriraj Hospital, Mahidol University, Bangkok 10700, Thailand; 2The Golden Jubilee Medical Center, Mahidol University, Nakhon Pathom 73170, Thailand; romyenj@gmail.com

**Keywords:** hemorrhoidectomy, postoperative pain, analgesics, anesthesia, pain, complication, hemorrhoids, review

## Abstract

*Background and Objectives*: Excisional hemorrhoidectomy is considered as a mainstay operation for high-grade hemorrhoids and complicated hemorrhoids. However, postoperative pain remains a challenging problem after hemorrhoidectomy. This systematic review aims to identify pharmacological and non-pharmacological interventions for reducing post-hemorrhoidectomy pain. *Materials and Methods*: The databases of Ovid MEDLINE, PubMed and EMBASE were systematically searched for randomized controlled trails (published in English language with full-text from 1981 to 30 September 2021) to include comparative studies examining post-hemorrhoidectomy pain as their primary outcomes between an intervention and another intervention (or a sham or placebo). *Results*: Some 157 studies were included in this review with additional information from 15 meta-analyses. Fundamentally, strategies to reduce post-hemorrhoidectomy pain were categorized into four groups: anesthetic methods, surgical techniques, intraoperative adjuncts, and postoperative interventions. In brief, local anesthesia-alone or combined with intravenous sedation was the most effective anesthetic method for excisional hemorrhoidectomy. Regarding surgical techniques, closed (Ferguson) hemorrhoidectomy performed with a vascular sealing device or an ultrasonic scalpel was recommended. Lateral internal anal sphincterotomy may be performed as a surgical adjunct to reduce post-hemorrhoidectomy pain, although it increased risks of anal incontinence. Chemical sphincterotomy (botulinum toxin, topical calcium channel blockers, and topical glyceryl trinitrate) was also efficacious in reducing postoperative pain. So were other topical agents such as anesthetic cream, 10% metronidazole ointment, and 10% sucralfate ointment. Postoperative administration of oral metronidazole, flavonoids, and laxatives was associated with a significant reduction in post-hemorrhoidectomy pain. *Conclusions*: This systematic review comprehensively covers evidence-based strategies to reduce pain after excisional hemorrhoidectomy. Areas for future research on this topic are also addressed at the end of this article.

## 1. Introduction

Hemorrhoids is the most common benign anal disease encountered by physicians and surgeons [1]. It has been estimated that a lifetime risk of hemorrhoids could be as high as 75% in the general population [2]. Although most hemorrhoids can be treated effectively by medication and/or office-based procedures, surgical treatment is indicated in high-grade hemorrhoids or complicated diseases [3,4,5].

A systematic review and network meta-analysis of various operations for hemorrhoids demonstrated that non-excisional surgeries, such as doppler-guided hemorrhoidal artery ligation and stapled hemorrhoidopexy, were less painful than excisional hemorrhoidectomy [6]. However, the latter had less recurrence and was associated with a lower cost of surgical instruments. Moreover, both internal and external components of hemorrhoids can be effectively removed by hemorrhoidectomy, which is reasonably easy to learn and perform in an elective or emergency setting [2]. As a result, hemorrhoidectomy is still regarded as the mainstay operation for advanced and/or complicated hemorrhoids although post-hemorrhoidectomy pain remains a challenging problem [7,8]. The incidence of moderate to severe pain following conventional hemorrhoidectomy reported in the literature could be as high as 65% [9]. We hypothesized that some perioperative measures, together with refined surgical and anesthetic techniques, could minimize pain after hemorrhoidectomy. This study, therefore, aimed to systematically review strategies to reduce post-hemorrhoidectomy pain published in the literature.

## 2. Materials and Methods

This systematic review was conducted in accordance with the Preferred Reporting Items for Systematic Reviews and Meta-Analyses (PRISMA) 2020 statement [10]. Literature published in English language with full text and indexed in Ovid MEDLINE, PubMed and EMBASE from 1981 to 30 September 2021 was independently searched by the two authors. The following search terms were used: (“hemorrhoidectomy” (Title) OR “haemorrhoidectomy” (Title)) AND (“postoperative pain” (All Fields) OR “posthemorrhoidectomy pain” (Title/Abstract) OR “posthaemorrhoidectomy pain” (Title/Abstract)). Synonyms of each of the terms were also used in the search. To obtain the highest level of scientific evidence, only randomized controlled trails (RCTs) and systematic reviews and meta-analyses of RCTs were included.

Fundamentally, comparative studies examining post-hemorrhoidectomy pain as their primary outcomes between an intervention and another intervention (or a sham or placebo) were included. Excisional hemorrhoidectomy was limited to open (Milligan-Morgan) technique, closed (Ferguson) technique, and semi-closed (modified Ferguson) technique. With this search strategy, 701 articles were eligible for further evaluation (168 from Ovid MEDLINE, 227 from PubMed and 306 EMBASE). References of the included studies were further reviewed to identify any additional suitable studies that may be missed by the aforementioned search strategy. Exclusion criteria included studies without extractable data, those with duplicate data, and those with sample size less than 20 patients. Finally, a total of 157 RCTs and 15 meta-analyses were included in this manuscript. Some of major guideline recommendations were also reviewed for additional information. PRISMA flow chart is shown in Figure 1.

It is worth noting that strategies to reduce pain after stapled hemorrhoidopexy, laser therapy, and radiofrequency ablation were not included in this review. Roles of conventional analgesics such as paracetamol and non-steroidal anti-inflammatory drugs on post-hemorrhoidectomy pain were not discussed in this review because the advantages of multimodal opioid-sparing analgesia are clearly evident in surgical practices including hemorrhoidectomy [11].

## 3. Results

Various strategies or interventions to reduce pain after excisional hemorrhoidectomy were identified and categorized into four groups: anesthetic methods, surgical techniques, intraoperative adjuncts, and postoperative interventions.

### 3.1. Anesthetic Methods

Hemorrhoidectomy could be performed under general anesthesia, spinal anesthesia, caudal block, nerve block, local anesthesia, and combined anesthesia. Preferred anesthetic techniques could vary among patients, surgeons, hospitals, and countries. Effects of anesthetic techniques on post-hemorrhoidectomy pain have been extensively examined in at least 18 RCTs with a total number of 1465 patients [12,13,14,15,16,17,18,19,20,21]. A recent systematic review and meta-analysis of seven RCTs comprising 440 patients undergoing excisional hemorrhoidectomy (222 patients with local anesthesia plus intravenous sedation, and 218 patients with spinal anesthesia) has found that local anesthesia combined with intravenous sedation had a significantly lower pain score at 6 h and 24 h after an operation–with mean difference of numerical pain rating scale −2.25 (95% CI −3.26 to −1.24) and −0.87 (95% CI −1.33 to −0.40), respectively [20]. Moreover, local anesthesia combined with intravenous sedation was associated with a remarkably lower risk of rescue analgesia (risk ratio (RR) = 0.18, 95% CI 0.06–0.53), urinary retention (RR = 0.17, 95% CI 0.07–0.37), and headache (RR = 0.09, 95% CI 0.03–0.33) compared with spinal anesthesia.

Another recent systematic review and meta-analysis of nine RCTs (six for local anesthesia versus regional anesthesia and the others for local anesthesia versus general anesthesia) comprising 727 patients also demonstrated that hemorrhoidectomy under local anesthesia was associated with a significant reduction in the intensity of postoperative pain and length of hospitalization compared with that under regional or general anesthesia [21]. Local anesthetic methods included pudendal nerve block [12], ischiorectal block [13], perianal block [14], posterior perineal block [15], and local anesthetic infiltration into the wound [16,17]. Regarding choices of local anesthetic agents, most investigators would prefer to use a long-acting drug such as ropivacaine, bupivacaine, and liposomal bupivacaine.

Regarding spinal anesthesia, it is worth noting that adding midazolam or morphine to bupivacaine in spinal anesthesia resulted in better pain control in the first 12–24 h after hemorrhoidectomy [18,19].

### 3.2. Surgical Techniques

#### 3.2.1. Closed versus Open Technique

Comparison of postoperative pain between closed (Ferguson) hemorrhoidectomy and open (Milligan-Morgan) hemorrhoidectomy has been examined in a randomized fashion since early 1990s [22]. Pain following the two techniques was found to be comparable in earlier studies [23,24], but several later studies indicated less postoperative pain in closed hemorrhoidectomy [25,26,27]. The advantage of closed hemorrhoidectomy on postoperative pain was confirmed in a recent systematic review and meta-analysis of 11 RCTs comprising 1326 patients (663 in closed hemorrhoidectomy and 663 in open hemorrhoidectomy), in which closed technique was associated with a modest but significant reduction in post-hemorrhoidectomy pain (standardized mean difference −0.36; 95%CI −0.64 to −0.07) [28].

#### 3.2.2. Scissors, Diathermy or Other Instruments

During excisional hemorrhoidectomy, a variety of surgical instruments have been used to remove hemorrhoidal tissue, including scissors, diathermy, laser, an ultrasonic scalpel, a vascular sealing device (or bipolar electrosurgical device), and a radiofrequency device. Two prospective randomized studies demonstrated comparable postoperative pain between scissors and diathermy for ‘conventional or traditional’ hemorrhoidectomy [29,30]. Meanwhile, several systematic reviews and meta-analyses demonstrated that hemorrhoidectomy with a vascular sealing device (Ligasure™) had significantly less postoperative pain, shorter operative time, and decreased blood loss compared with conventional hemorrhoidectomy [31,32,33]. Likewise, a recent several systematic review and meta-analysis of eight RCTs comprising 468 patients (233 in the ultrasonic scalpel group) has shown that hemorrhoidectomy with an ultrasonic scalpel (Harmonic^®^, San Jose, CA, USA) had advantages over conventional hemorrhoidectomy in terms of reduced postoperative pain and faster recovery [34].

#### 3.2.3. Hemorrhoidectomy Combined with Lateral Internal Anal Sphincterotomy

Since the spasm of the internal anal sphincter (IAS) was thought to be an aggravating factor for post-hemorrhoidectomy pain, chemical sphincterotomy, and lateral internal anal sphincterotomy (LIS) have been proposed to relieve postoperative pain. A recent systematic review of 2180 patients undergoing open or closed hemorrhoidectomy (about 43% having combined hemorrhoidectomy with LIS) has demonstrated less postoperative pain in those with LIS. However, there was a significant higher rate of fecal incontinence in patients with LIS compared with those without (7.7% versus 1.25%), although the severity of fecal incontinence was mild (e.g., flatus incontinence and fecal soiling) and improved over time [35]. In 2021, a single-institute RCT of 200 patients from India also confirmed the efficacy of LIS on better pain relief after hemorrhoidectomy, without compromising anal continence [36].

### 3.3. Intraoperative Adjuncts

#### 3.3.1. Injection of Botulinum Toxin

Following the hypothesis of IAS-spasm induced post-hemorrhoidectomy pain, injection of botulinum toxin A into the IAS has been used to induce a transient relaxation of the IAS–instead of surgical division of the IAS which might cause long-term sequelae such as fecal incontinence. The effect of botulinum toxin injection on post-hemorrhoidectomy pain has been examined in a few RCTs, with conflicting results [37,38,39]. Although the injection of botulinum toxin reduced maximal resting pressure and maximal squeeze pressure up to 12 weeks after hemorrhoidectomy in these RCTs, only two out of the three studies reported a significant reduction in post-hemorrhoidectomy pain in patients treated with an injection of botulinum toxin [37,38]. Moreover, an intraoperative injection of botulinum toxin was shown to be more effective than repeated applications of glyceryl trinitrate in decreasing pain after hemorrhoidectomy [40].

#### 3.3.2. Intradermal Injection of Methylene Blue

Methylene blue has a unique analgesic activity by temporarily interfering sensory nerve conduction at the nerve endings (pain and itch receptors) within the epidermis and dermis. Intradermal injection of 1% methylene blue 4 mL at the site of open hemorrhoidectomy was shown to reduce pain in the first three postoperative days, without increasing any complication [41].

#### 3.3.3. Intrasphincteric Injection of Ketorolac

A prospective study was conducted in 1994 to compare the postoperative analgesic effect of between ketorolac injected into the anal sphincter muscle at the time of hemorrhoidectomy and taken orally thereafter versus standardized narcotic intramuscular/oral analgesics. The authors reported comparable pain intensity between the two groups—the ketorolac group had a higher satisfaction rating—with facilitating early discharge in the setting of ambulatory hemorrhoidectomy [42].

### 3.4. Postoperative Interventions

#### 3.4.1. Topical Calcium Channel Blockers and Glyceryl Trinitrate

Aiming to decrease the spasm of IAS, calcium channel blockers (diltiazem or nifedipine) and glyceryl trinitrate (GTN) have been introduced as a topical agent applied into the anal canal and/or onto the perianal skin as an intervention to improve post-hemorrhoidectomy pain. A systematic review and meta-analysis of 12 RCTs with 1095 patients found that the topical application of GTN was associated with a significant pain reduction up to 2 weeks after hemorrhoidectomy and a faster rate of wound healing (by 4–10 days) comparing with a placebo. However, 10% of patients treated by GTN experienced headache, which could limit the extensive use of such an agent [41]. Similarly, a meta-analysis of five RCTs (227 patents) examining the effect of topical calcium channel blockers on post-hemorrhoidectomy pain indicated a significant pain reduction on postoperative day 1–4 (pooled mean difference in degree of pain score of approximately −3.5) in those treated by diltiazem ointment. Of note, there was no significant difference in the incidence of headache between diltiazem and placebo group [43].

#### 3.4.2. Topical Anesthetic Cream

The application of topical anesthetic cream (EMLA™ (Eutectic Mixture of Local Anesthetics) cream; mixture of 2.5% lidocaine and 2.5% prilocaine) as a preemptive analgesia to reduce pain during and after hemorrhoidectomy has been studied since 2000 [44]. Initially, it was applied over the perianal skin before infiltrating local anesthesia (perianal block) for hemorrhoidectomy. Later, the beneficial effects of topical EMLA™ cream on post-hemorrhoidectomy pain were determined when the cream was applied immediately after hemorrhoidectomy (either within the anal canal or over the perianal skin). Two double-blind RCTs demonstrated that topical EMLA cream significantly reduced pain intensity in the first 2–24 h after hemorrhoidectomy, without any adverse drug reactions [45,46].

#### 3.4.3. Other Topical Medications

Metronidazole: Metronidazole has antibacterial activity against enteric anaerobes (which could exert inflammatory pain after hemorrhoidectomy) and antioxidant effect (which could minimize pain and promote wound healing) [47]. A systematic review and meta-analysis of four RCTs including 149 patients (76 with 10% metronidazole ointment and the others with placebo) demonstrated that topical metronidazole significantly reduced post-hemorrhoidectomy pain, with an approximate mean difference of −2 to −1 in visual analog scale for pain throughout the first two weeks postoperatively [48]. There was no serious adverse drug reaction reported in patients treated by topical metronidazole although some experienced perianal burning and itching.

Sucralfate: Sucralfate, a complex of aluminum hydroxide and sucrose octasulfate, acts as a mucosal protective barrier and a promotor for mucosal healing [49]. It is approved by the U.S. Food and Drug Administration for the treatment of duodenal ulcers. Due to its unique mucoprotective effect of sucralfate, its topical form (10% sucralfate ointment) has been investigated whether it can reduce pain after open hemorrhoidectomy in at least three RCTs [50,51,52]. Applying ointment onto a hemorrhoidectomy wound once or twice daily for two weeks, topical sucralfate was shown to significantly reduce post-hemorrhoidectomy pain throughout the period of drug application and shorten time to wound healing compared with a placebo ointment.

Diclofenac: The application of diclofenac rectal suppository provided a better pain control in the first 24 h after hemorrhoidectomy compared with a placebo [45]. Notably, the analgesic effect of this topical non-steroidal anti-inflammatory drug lasted longer than that of EMLA™ cream.

Baclofen: Baclofen is a gamma-amino butyric acid (GABA) receptor agonist. It has been used traditionally as a muscle relaxant and a medication for neuropathic pain. However, it was postulated that post-hemorrhoidectomy pain could be a result of spasticity of the anal sphincter complex and injury to sensory nerves at the anoderm [53]. As a result, baclofen could be a new analgesia for controlling post-hemorrhoidectomy pain. In fact, a recent small double-blind RCT demonstrated the beneficial effect of 5% baclofen cream on pain after open hemorrhoidectomy. In this study, baclofen cream was immediately applied after surgery and then every 12 h for 2 weeks, and its analgesic effect was clearly evident after the first few days after the procedure [54].

Cholestyramine: Since some investigators believed that bile acids in the stool could cause skin irritation and inflammation in the perianal area as noted in an etiology peristomal dermatitis, they conducted a small double-blind RCT comprising 91 patients undergoing open hemorrhoidectomy [55]. Either 15% cholestyramine ointment or placebo was applied on perianal skin (but not inside the anus) immediately after surgery and then every 8 h for 2 weeks. Patients treated by cholestyramine ointment had a significant lower intensity of pain only in the first 48 h after an operation.

Aloe vera: Aloe vera was found to have anti-inflammatory activity and promote cutaneous healing [56]. The effects of Aloe vera cream on pain and wound healing after hemorrhoidectomy have been investigated in a few, small RCTs with conflicting results [57,58].

Vitamin E: The anti-inflammatory effect of vitamin E in dermatologic diseases was investigated in the management of post-hemorrhoidectomy pain. A small double-blind RCT in Spain included 60 patients undergoing open hemorrhoidectomy (30 with vitamin E ointment and the others with petrolatum as a placebo) [59]. Two mL of ointment was applied onto a hemorrhoidectomy wound twice daily for one week. The investigators found that vitamin E ointment significantly reduced post-hemorrhoidectomy pain–approximately with a mean difference of −4 in visual analog scale for pain throughout the period of the ointment applied. Undoubtedly, more studies with larger sample sizes need to be conducted to confirm these findings.

Trimebutine: As an antispasmodic used orally to treat intestinal cramping and irritable bowel syndrome, Trimebutine (Proctolog^®^, Pfizer, New York, NY, USA) was used topically to treating anal fissures. In a RCT including 160 hemorrhoidectomies, a trimebutine suppository significantly reduced resting anal pressure but had no effect on post-hemorrhoidectomy pain [60].

#### 3.4.4. Oral Metronidazole

With its properties of anti-aerobic activity and antioxidant effect, oral metronidazole is often used to prevent surgical site infection and minimized pain following hemorrhoidectomy [47]. In the last five years, there have been at least three systematic reviews and meta-analyses of maximum nine RCTs (523 patients) examining the analgesic effect of oral metronidazole on post-hemorrhoidectomy pain [47,48,61]. With some limitations of small sample-size studies and heterogeneity among studies, these meta-analyses demonstrated that the intensity of postoperative pain at all time points of measurement (up to a week postoperatively) in patients receiving oral metronidazole was significantly less than that in comparison groups. It appeared that the analgesic effect of oral metronidazole on post-hemorrhoidectomy was slightly prominent than that of topical metronidazole [48], although there is no RCT comparing this effect between the two routes of administration. Pain intensity during the second and third week after hemorrhoidectomy between a seven-day postoperative administration of oral metronidazole and placebo was examined in a recent small RCT from Australia, in which the investigators found no significant difference in median worst pain scores and defecation-related pain [62].

#### 3.4.5. Flavonoids

Flavonoids, known as a venoactive drug, was shown to increase venous tone, decrease vascular permeability, improve lymphatic drainage, and reduced microvascular and tissue inflammation [63]. It has been used effectively in the treatment of low-grade hemorrhoids as well as chronic venous insufficiency [3]. As it has activities against local inflammation in hemorrhoidal tissue, the effects of flavonoids on post-hemorrhoidectomy pain have been examined in several RCTs [64,65,66].

Systematic reviews and meta-analyses of three RCTs including 216 patients showed that postoperative administration of flavonoids reduced pain intensity after hemorrhoidectomy by approximately 1 out of 10 in visual analog scale and postoperative analgesic requirement by 46% [67,68].

#### 3.4.6. Laxatives

Bowel confinement offered no benefits in clinical and patient-reported outcomes after various anorectal operations [69]. In contrast, postoperative use of laxatives including bulk-forming agents and osmotic laxatives was associated with earlier and less painful defecation following anal surgeries including excisional hemorrhoidectomy [70,71,72].

#### 3.4.7. Mesoglycan

Mesoglycan is a porcine-derived polysaccharide complex with antithrombotic and profibrinolytic properties. It was shown to regulate selective permeability at the microcirculatory level and restore microvascular injury [73]; thus, potentially reducing edema and pain at a surgical site. Since hemorrhoids were associated with hypervascularity and higher blood flow of the anal cushions [7], the effects of mesoglycan on post-hemorrhoidectomy pain were examined first as a pilot prospective multicenter study in Italy comprising 101 hemorrhoidectomies. The mesoglycan-treated group received mesoglycan 60 mg intramuscularly once daily for the first 5 days postoperative, followed by 50 mg (one tablet) orally twice daily for 30 days. The investigators found that patients treated by mesoglycan had less pain at 7–10 days after surgery compared with patients with no drug treatment [74]. Later, a large multicenter observational study (398 patients) supported by Italian Society of Colorectal Surgery also found that patients treated by the aforementioned course of mesoglycan experienced less post-hemorrhoidectomy pain (from the first day to 6 weeks after surgery) and earlier return to their normal activities-compared with historical control individuals [75].

#### 3.4.8. Warm Sitz Bath

Physiological studies showed that anal resting pressure diminished significantly after a warm sitz bath (40 Celsius) for 5–10 min [76]. The effects of warm water on the relaxation of IAS could last up to 70 min after exiting the bath [77]. Although a warm sitz bath is commonly advised to patients with anorectal pain and those undergoing anorectal operations [78], several systematic reviews did not find its benefit on reducing pain or enhancing wound healing in various anal disorders [79,80,81]. With regard to post-hemorrhoidectomy pain, a warm sitz bath did not provide better pain relief [82]. However, a small RCT showed that using surgical glove filled with warm water applied to the perianal area four times a day reduced pain in the first few days after hemorrhoidectomy [83].

#### 3.4.9. Avoidance of Spicy Foods

Capsasin, a chemical causing spiciness in chili, elicits burning pain by activating vanilloid receptors on sensory nerve endings in human skin and mucosa [84]. A small RCT from India showed that patients consuming 3 g of chili powder per day during the first week after hemorrhoidectomy had a significantly higher degree of anal burning and pain intensity than those without chili consumption [85].

#### 3.4.10. Transcutaneous Electrical Nerve Stimulation and Acupuncture

In the literature, alternative therapies for managing post-hemorrhoidectomy pain included transcutaneous electrical nerve stimulation (TENS) and acupuncture. These non-pharmacological interventions aim to modulate neurotransmitters along the nociceptive pathway to reduce hyperalgesia [86]. TENS on the dorsal web between the first and second metacarpal bone and on the radial side of forearm was shown to reduce pain and opioid consumption after hemorrhoidectomy [87]. A recent large network meta-analysis of 101 RCTs (10,972 hemorrhoidectomies—mostly from China) confirmed the efficacy of acupuncture and its related techniques in post-hemorrhoidectomy pain relief [88]. A systematic review and network meta-analysis of RCTs evaluating all somatosensory stimulation treatments for post-hemorrhoidectomy pain is still underway [89].

#### 3.4.11. Patient’s Checklist for Analgesic Consumption

High compliance with prescribed analgesic consumption would improve postoperative pain control. The benefits of medication checklist for patients taking analgesia following hemorrhoidectomy were examined in a small RCT, in which individuals in the self-checklist group had a minimal, but statistically significant, reduction in average postoperative pain between day 1 and day 14 compared with those in the control group (−2.51 versus −1.86 in visual analog scale) [90]. Nevertheless, this non-pharmacological adjunct appeared to be useful, at no cost for reducing post-hemorrhoidectomy pain especially in individuals with polypharmacy.

### 3.5. Limitations

Although this systematic review addressed strategies to reduce post-hemorrhoidectomy pain using a systematic search of three major biomedical literature databases (Ovid MEDLINE, PubMed and EMBASE), it still has several limitations. First, other electronic sources such as Cochrane Review Library and Cumulative Index to Nursing and Allied Health Literature (CINAHL) are not included in this review. Second, only English-language articles with full text were included in this searching strategy. Third, although all studies included in this review were RCTs, their quality and number of sample size were various. Therefore, it is difficult to compare the treatment effects among these interventions. Fourth, detailed data are lacking for some interventions such as local anesthetic technique and intraoperative botulinum toxin injection.

### 3.6. Areas for Future Research

In the era of enhanced recovery after surgery, the management of post-hemorrhoidectomy pain must include evidence-based preoperative, intraoperative, and postoperative interventions. The Prospect (procedure specific postoperative pain management) launched the recommendations for pain management after hemorrhoidectomy in 2010 and updated the recommendations in 2017 [91,92]. It is worth noting that these recommendations are mostly derived from individual interventional studies rather than multimodal studies. Interesting, recent evidence has supported a multimodal approach to post-hemorrhoidectomy pain, e.g., a combination of flavonoids and oral metronidazole had a greater degree of pain reduction after hemorrhoidectomy than either single medication [93]. Moreover, increased adherence to guideline recommendations or surgical care bundles was associated with better short-term and long-term surgical outcomes [94,95]. Whether multimodal approaches to post-hemorrhoidectomy pain and their adherence would result in better pain relief needs to be investigated. Within a care bundle for reducing post-hemorrhoidectomy pain, it is also interesting to determine which interventions have a great impact on postoperative pain. New pharmacological and non-pharmacological measures are still required to reduce post-hemorrhoidectomy pain. Moreover, the effect of surgeon’s experience on pain after hemorrhoidectomy remains unknown.

## 4. Conclusions

Postoperative pain remains an unsolved and disturbing problem after excisional hemorrhoidectomy. Although there are variations in the methodology and quality of the included studies, strategies to reduce post-hemorrhoidectomy pain are presented in this systematic review and summarized in Table 1.

## Figures and Tables

**Figure 1 medicina-58-00418-f001:**
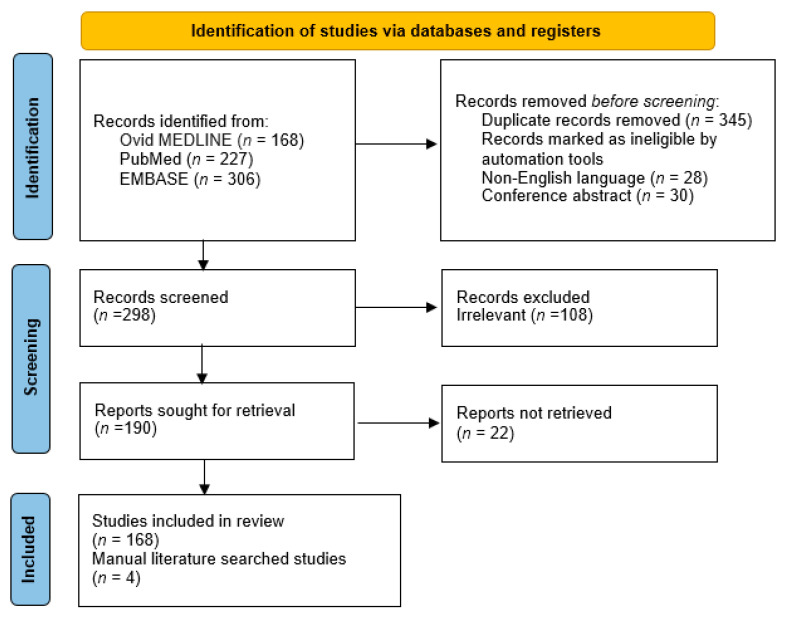
PRISMA flow chart.

**Table 1 medicina-58-00418-t001:** Strategies to reduce post-hemorrhoidectomy pain.

Anesthetic methods
Local anesthesia, alone or combined with intravenous sedation (the most preferred method), spinal anesthesia and general anesthesia Adding midazolam ^1^ or morphine ^1^ to bupivacaine in spinal anesthesia
Surgical techniques
Closed (Ferguson) hemorrhoidectomy Hemorrhoidectomy with a vascular sealing device or an ultrasonic scalpel Hemorrhoidectomy combined with lateral internal anal sphincterotomy
Intraoperative adjunct
Intrasphincteric injection of botulinum toxin (±) Intradermal injection of methylene blue ^1^ Intrasphincteric injection of ketorolac ^1^
Postoperative interventions
Topical agents: calcium channel blockers, glyceryl trinitrate, anesthetic cream, metronidazole, sucralfate, baclofen ^1^, cholestyramine ^1^, trimebutine ^1^, vitamin E ^1^, diclofenac ^1^, Aloe vera (±) Oral metronidazole Flavonoids Laxatives Mesoglycan ^1^ Avoidance of spicy foods ^1^ Transcutaneous electrical nerve stimulation Acupuncture Checklist for analgesic consumption ^1^

Notes: ± = conflicting results, ^1^ = only single randomized controlled trial was identified.

## Data Availability

All data analyzed during this study are included in this published article.

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
