# Peer review of "Strategies to Reduce Post-Hemorrhoidectomy Pain: A Systematic Review"

_medicina, 2022, doi:10.3390/medicina58030418_

Round 1
Reviewer 1 Report
What was the limitations of review?
interpretation of results is not sufficient
Author Response
Thank you very much for your kind comments.
According to your kind suggestion, we address the limitation of this review before the end of the revised manuscript.
Kindly be noted that we originally aimed to write a narrative review with a systematic search (to achieve a comprehensive and broad review) but wrongly addressed it as a systematic review. We therefore delete the word 'systematic' from the title of manuscript. Although we did not interpret the results, we summarized the results of 'strategies to reduce post-hemorrhoidectomy pain" in one table and addressed area of future researches towards the end of the manuscript.
Reviewer 2 Report
Firstly, I would like to congratulate the authors for a well-constructed review. The review has merit and will be of interest to our readers. However, it needs to be revised before it can be reconsidered. My comments are appended below:
Abstract: Well-written. No changes are needed.
Introduction: What was your hypothesis before conducting this meta-analysis? Please mention in one or two lines at the end of the Introduction section.
Methods: The search strategy is not optimal. For an optimal database search, please see the study below. At least four databases need to be searched for an appropriate search strategy.
https://systematicreviewsjournal.biomedcentral.com/articles/10.1186/s13643-017-0644-y
-Please mention the detailed search strategy. How was the search done? In cases of any disagreements, how was the consensus made?
-How was the quality assessment of the included studies done? Did you use any objective assessment tools?
Results: The authors have included results about previous systematic reviews and meta-analyses? Please exclude the results of already published reviews and meta-analyses.
-Please mention the characteristics of the included studies? What were the levels of evidence of these studies?
Author Response
Thank you very much for your kind comments and suggestion. We do agree with you that we would have searched more biomedical electronic databases. As a result, we additionally searched Ovid MEDLINE and noted it in the materials and methods section. However, since the data search was done in only 3 major data sources, we have addressed it as one of our limitations at the end of the revised manuscript.
Kindly be noted that we originally aimed to write a narrative review with a systematic search (to achieve a comprehensive and broad review) but we wrongly addressed it as a systematic review. Therefore, we removed the word 'systematic' from the title of the manuscript. Detailed data search was also re-written in the materials and methods section in order to make readers appreciate the searching strategy (term, inclusion criteria and exclusion criteria) and its results in a snapshot.
Notably, no meta-analysis was conducted in this narrative review. On the other hand, some key meta-analyses published in the literature were mentioned in this review to show readers a treatment effect of an intervention.
According to your valuable suggestion, we added our hypothesis at the end of the introduction part. However, we did not use a formal objective tool for assessing the quality of RCTs included in this review. Therefore, we also addressed this drawback as one of our limitations at the end of the revised manuscript.
Some typo-error and grammar errors have been corrected.
Reviewer 3 Report
This is a review of RCTs and systematic reviews and meta-analyses of RCTs on pain after hemorrhoidectomies. It is an ambitious effort to collect and go through all these studies. However, I have some suggestions, comments and questions about the presentation with my main concern being that there is no discussion section.
If meta-analyses are referred to it would be helpful to know the number of RTCs and the total number of patients included in that study. If there are several meta-analyses on the same topic it is also of interested to know how the included studies overlap. It is also important to know if the authors found RTCs on the same topic as the meta-analyses they referred to that are not included in the meta-analyses.
Under 3.1 the number of studies found describing these topics could be explained better. For example, how many studies where found in total describing these topics? Where there any other studies found except those included in the meta-analyses?
It is mentioned that local anesthesia can be used in conjunction with spinal anesthesia, are there any studies on this topic and why are not other combinations such as general anesthesia in combination with local anesthesia mentioned?
Under 3.1, how can 222 patients in one group and 218 in the other add up to 400 total?
What is meant by intravenous anesthesia, is it total intravenous (general) anesthesia or is it intravenous sedation?
Words such as interestingly is better suited for the discussion than for the results section. An explanation why the authors find something interesting is also needed.
The paragraph under 3.2.2 that starts with “It is plausible that…” is better suited for the discussion than for the results section.
How does the authors interpret the difference in results for the studies on intraoperative botulinum toxin injections? This could be further explored.
In the first half of 3.4.3. the substances discussed are in italics but not in the later half.
The section on flavonoids refers both to RCTs and systematic reviews but it is difficult to understand if the RCTs are part of the systematic reviews or if they reflect some other aspect of flavonoid treatment.
The section 3.8 is better suited for the discussion than for the results section.
Author Response
Thank you very much for your kind comments.
Your valuable suggestion on noting a number of RCTs and patients included in meta-analyses mentioned in this review is highly appreciated. We have done accordingly.
As you clearly point-out, we avoided to address several meta-analyses on the same topic because most of the time they contained overlapping studies. In fact, we selected either the largest or the most recent meta-analysis to be included in this review (to give readers a treatment effect of an intervention).
We do agree with you that it is important to know and mention if newer RCTs on the same topic as the meta-analyses they referred to that are not included in the meta-analyses - particularly the outcomes of a newer RCT have contradicted those of meta-analysis. Since we included 15 meta-analyses in this review, only one newer RCT (Vijayaraghavalu, 2021) was identified in the topic of 'Hemorrhoidectomy combined with lateral internal anal sphincterotomy'. We have mentioned this article in that section based on your kind suggestion.
Thank to your valuable comments on section 3.1 (anesthetic technique), we have revised our manuscript accordingly - including a correction of number typo error (from a wrong 400 to a correct 440), a removal of statement mentioning about local anesthesia can be used in conjunction with spinal anesthesia and general anesthesia (due to a lack of high-quality data supported), and a removal of the word 'intravenous anesthesia' (to avoid confusion between total intravenous general anesthesia and intravenous sedation).
We also do agree with you that the word 'interestingly' should not be included in the result part. Therefore, we deleted this word from the revised manuscript. Also paragraph starting with “It is plausible that…” was removed because it is an explanation of the result rather than the results per se.
As you precisely mentioned, it could be difficult to interpret the difference in results for the studies on intraoperative botulinum toxin injections because the injection technique and dosage may be various from one study to another study. Therefore, we addressed this caution in our study limitation.
Typing style (italics vs non-italics) was corrected according to your kind notification.
In the section on flavonoids, since the two systematic reviews and meta-analyses included 3 RCTs mentioned in previous paragraph and reflected the same aspect of flavonoid treatment, we added information of these three RCTs with a total number of cases studied in the revised manuscript.
Kindly be noted that we originally aimed to write a narrative review with a systematic search (to achieve a comprehensive and broad review) but wrongly addressed it as a systematic review. We therefore delete the word 'systematic' from the title of manuscript. Although we did not interpret the results, we summarized the results of 'strategies to reduce post-hemorrhoidectomy pain" in one table and addressed area of future researches towards the end of the manuscript.
Round 2
Reviewer 2 Report
The authors have addressed all my comments in the revised manuscript. The revised manuscript looks much better. I would like to congratulate them for their work.
Author Response
We are most grateful to your kind comments and suggestions. Once again thank you very much.
Reviewer 3 Report
I find that the authors have done a nice revision of their work that clarifies my earlier questions. They have also corrected the errors I have pointed out. I do not have anything to add to this work at this point.
Author Response

(The authors gave the same response as above.)
